# Effect of Preheating Treatment before Defatting on the Flavor Quality of Skim Milk

**DOI:** 10.3390/molecules24152824

**Published:** 2019-08-02

**Authors:** Lingjun Tong, Huaxi Yi, Jing Wang, Minghui Pan, Xuelu Chi, Haining Hao, Nasi Ai

**Affiliations:** 1College of Food Science and Engineering, Ocean University of China, 5th Yushan Road, Qingdao 266003, China; 2Beijing Advanced Innovation Center for Food Nutrition and Human Health, Beijing Engineering and Technology Research Center of Food Additives, Beijing Higher Institution Engineering Research Center of Food Additives and Ingredients, Beijing Technology & Business University, Beijing 100048, China

**Keywords:** skim milk, preheating treatment, sensory analysis, e-nose, volatile compounds

## Abstract

Skim milk has a poor flavor due to the lack of fat. Finding ways to improve the flavor quality of skim milk has attracted the attention of more and more researchers. The purpose of this study was to create a skim milk product with good flavor by processing. Briefly, raw milk was treated by preheating at pasteurization (85 °C, 15 s) and ultra-high temperature (UHT) instantaneous sterilization (137–141 °C, 4 s). Subsequently, the sample was centrifuged to remove fat and obtain two kinds of skim milk, namely, PSM (skim milk obtained by preheating at 85 °C, 15 s) and USM (skim milk obtained by preheating at 137–141 °C, 4 s). The results showed that the intensity of the main sensory attributes (overall liking, milk aroma, etc.) and the concentrations of the key flavor compounds (2-heptanone, 2-nonanone, decanal, hexanoic acid, etc.) were significantly higher in the USM (*p* < 0.05) than that of the PSM and RSM (skim milk without preheating). Principal component analysis (PCA) with E-Nose (electronic nose) showed that the RSM had significant differences in the milk aroma compared with the PSM and USM. Furthermore, it was found that there were good relationships between volatile compounds and sensory attributes by partial least squares regression (PLSR) analysis. These findings provided insights into improving the flavor quality of skim milk by preheating treatment instead of any flavor additives.

## 1. Introduction

Milk is a valuable nutritious food, which contains proteins, fat, sugar, amino acids, minerals, vitamins, and other ingredients required by the human body. It has been considered one of the perfect foods in the human diet. Milk consumption is always listed as an important element in a healthy and balanced diet [1]. However, with the improvement of the living standard, people (especially obese people) have become more sensitive to foods that contain fat. Studies show that a high-fat diet increases the amount of fat in the serum and liver, leading to intestinal dysfunction and impaired intestinal barrier function [2]. Therefore, skim milk is often preferred by consumers due to the loss of fat. Additionally, some studies demonstrate that the intake of skim milk is considered more beneficial than whole milk [3]. Nevertheless, loss of fat can lead to skim milk’s poor flavor. Previous studies have shown the influence of milk fat content on flavor release in dairy desserts [4,5]. McCarthy et al. pointed out that the consumer’s preference for milk increased with increasing fat content [6]. Therefore, it is particularly important to improve the flavor of skim milk.

Milk fat is known to have the most direct contribution to sensory aromas and flavor perception [7,8]. Milk contains numerous lipids, such as triacylglycerols, diacylglycerides, saturated/polyunsaturated fatty acids, and phospholipids. Heat treatment, as an important step in the milk process, can induce lipid oxidation and produce free fatty acids, methyl ketones, and oxidized fatty acids, which are directly correlated with milk flavor [9,10]. In heated (at or above 72 °C) milks, some lactones, methyl ketones, and aldehydes are formed from milk fat. Some carbonyl compounds that have also been identified in heated milk may be derived from milk fat [11]. Notably, Gathercole et al. pointed out that in the presence of lipids, the concentrations of free fatty acids, methyl ketones, and oxidized fatty acids were directly related to the heat treatment temperature [9]. In addition, compared with ultra-pasteurized (UP)-treated (about 138 °C, 2 s) skim milk, UP-treated 2% fat milk had better sensory properties (overall aroma, sweetly aromatic, etc.) and a higher content of lipid-oxidized flavor compounds (hexanal, 2-heptanone, etc.) [12]. 

In general, in the presence of lipids, heat treatment (at or above 60 °C) can also promote the formation of milk fat globule membrane (MFGM) fragments or induce direct interactions with casein or serum proteins [13,14]. Moreover, the interaction of these milk matrices due to thermal treatment can lead to the changes in taste and aroma of milk [10,15]. This may be attributed to the fact that the lysine residues of proteins can react with the milk’s reducing carbohydrates, resulting in the so-called non-enzymatic browning or Maillard reaction during heat treatment and inducing both protein and lipid modification [9,16]. Taken together, these reactions affect the sensory properties (aroma, flavor, and appearance) of milk by heat treatment in the presence of fat. However, raw milk is usually directly centrifuged (or preheated at a lower temperature, about 50 °C) to remove fat, and then heat-treated, whereby skim milk is unable to obtain a good flavor brought about by fat. Notably, the level of most oxidized fatty acids and the concentration of methyl ketones (mainly 2-heptanone and 2-pentanone, the key flavor components in the milk [17]) were increased with temperature in the presence of fat [18]. Therefore, we assume that in the presence of fat, the milk is preheating treated (at a higher temperature) and then centrifuged to remove fat, which produces better flavored skim milk.

Currently, reports on the improvement of skim milk flavor by changing the process parameters are very rare. In our previous study, we improved the skim milk flavor profiles by lipase-catalyzed reactions and preheating at a lower temperature [19,20]. In this study, raw milk under different preheating treatments (85 °C, 15 s and 137–141 °C, 4 s) was centrifuged to create skim milk samples of satisfying milk flavor. The key flavor compounds in skim milk were monitored by GC/MS analysis. E-nose analysis and sensory evaluation were conducted to estimate and compare the flavor changes of skim milk by different preheating treatments.

## 2. Results and Discussion

### 2.1. Sensory Descriptive Analysis (DA)

The sensory profiles of three different samples are described in Table 1. The ANOVA analysis for the individual descriptors indicated that 6 of the 12 descriptors showed significant differences among the products. Specifically, it was observed that the sensory score of 7 descriptors (overall liking, milk aroma, cooked/sulfurous, caramelized, milk flavor, sweet taste, and residual mouthfeel) was increased with the increase of preheating temperature, whereas the score of fishy and salty was reduced gradually. It was reported that raw milk was characterized by a low but distinct salty taste and by grassy and mothball flavors [21]. It showed that preheating treatment could remove the raw milk’s bad flavor. Compared with the PSM (skim milk obtained by preheating at 85 °C, 15 s), RSM (skim milk without preheating) and W50/T30 (skim milk obtained by preheating at 50 °C, 30 min) [20], the intensity of overall liking, milk aroma, caramelized, milk flavor, and residual mouthfeel was closer to the maximum of the hedonic scale in skim milk after preheating treatment at 137–141 °C. Cooked/sulfurous and sweet taste intensity increased marginally at a higher temperature. The potential sources of these flavors were often attributed to Maillard and lipolysis reaction products [22]. In addition, lipid oxidation may also produce “oxidized” flavors caused by the formation of ketones and aldehydes [23]. It was consistent with the result of the GC-MS analysis. The aldehydes and ketones were increased significantly (*p* < 0.05) in the preheating treatment at 137–141 °C (Table 2). There was no difference in the intensity of other sensory attributes (e.g., umami, sour, and astringency) under different conditions (*p* > 0.05). It was demonstrated that these sensory attributes may not give rise to flavor changes of samples under different heat treatment conditions.

Sensory descriptive analysis was the most effective means of evaluating the milk’s flavor quality. It can truly reflect the wishes of consumers. Compared with the sensory evaluation analysis of the previous study [20], the sensory descriptive analysis was further discussed by cluster analysis (CA). CA was widely used to identify groups of similar characteristics [24]. In this study, cluster analysis was used to show sensory attribute differences between the three samples. As shown in Figure 1, the differences were present in the sensory characteristics of the three samples. Among them, the RSM and PSM were classified as a cluster and the USM (skim milk obtained by preheating at 137–141 °C, 4 s) was divided into a separate cluster. The cluster analysis clarified the fact that the sensory characteristics of skim milk were not changed significantly at pre-pasteurization temperature, whereas the preheating treatment at 137–141 °C could cause a great change in terms of the milk’s sensory characteristics. This change may be due to lipid oxidation and Maillard reaction caused by the ultra-high temperature (UHT) preheating treatment in the presence of fat [25]. Meanwhile, the analysis showed that, compared with the sensory quality of preheating treatment at 137–141 °C, it did not change much in skim milk obtained by preheating treatment at low temperatures.

### 2.2. E-Nose Response to Milk Aroma

The complexity of the milk matrix can affect the release of aroma in the milk. The difference in the aroma of skim milk was difficult to distinguish by sensory evaluation personnel. Therefore, based on the results of the previous study [20], E-nose was used to further analyze the skim milk’s aroma. The milk aroma was detected by E-nose with typical response signals from ten sensors. The data were used for the principal component analysis (PCA). PCA is a way to identify data patterns and can reduce the dimensionality of initial data and highlight differences and similarities of analyzed samples [26]. The PCA plot with E-nose is shown in Figure 2. The results showed that the total contribution variance of PC1 and PC2 was over 99%. There was a distinct separation among the RSM, PSM, and USM. The first principal component factor allowed the separation of the RSM, the PSM in the positive region, and the USM in the negative region, which indicated that the RSM and PSM had significantly different aromas compared with the USM. The second factor separated the RSM with the PSM and USM, which meant that the RSM had differences in the aroma with the PSM and USM. Overall, the RSM had significant differences in the milk aroma with the PSM and USM. Preheating treatment of milk causes the formation of volatile compounds from milk proteins, carbohydrates, and lipids, as well as some other compounds, which can affect the aroma of skim milk. Korhonen et al. indicated that heat treatment can induce lipid modification, notably oxidation, which affects aroma and flavor [16].

### 2.3. Analysis of the Volatiles in the RSM, PSM, and USM by Headspace Solid-Phase Microextraction (HS-SPME)-GC-MS

As can be seen from Figure 3a, the polydimethylsiloxane/divinylbenzene (PDMS/DVB) presented the highest sensitivity for volatile compounds of the milk. It could absorb nine types of volatiles, and the total concentrations of volatile compounds extracted were significantly higher than the other fibers. Therefore, the PDMS/DVB fiber was used to extract the volatile compounds in the milk samples.

The volatile compounds were isolated from three kinds of skim milk samples using SPME-GC-MS. The difference in the total ion currency (TIC) profile of the volatiles from the skim milk samples are shown in Figure 3b. The type and content of flavor compounds of the RSM, PSM, and USM were analyzed (Table 2). Based on the qualitative and quantitative analysis of GC-MS, a total of 14 volatile compounds was found in the three skim milk samples, which were identified in previous reports [19,27]. As shown in Table 2, it was observed that the types of volatile compounds had significant differences among the three samples. Numerically, 7 volatile compounds were identified in the RSM and PSM, and 10 volatile compounds were extracted from the USM. In terms of concentration, the total concentration of volatiles in the USM was almost ten times higher than that of the RSM and PSM. Moreover, a heat map was also created to show the difference in volatile compositions among the three kinds of skim milk samples (Figure 4). The USM sample contained more volatiles in higher concentrations, whereas the PSM and RSM sample had less volatiles in higher concentration. On the other hand, the samples were grouped into two clusters. The RSM and PSM were classified as a cluster and the USM was divided into a separate cluster, which was similar to the CA result of the sensory evaluation. 

Our previous study addressed a pretreatment at low temperature [20]. Compared with the low temperatures (30–60 °C), we found in this study that the concentrations of the key flavor compounds (2-heptanone, 2-nonanone, octanal, hexanoic acid, etc.) were significantly higher in the skim milk obtained under the high-temperature (137–141 °C) treatment. Specifically, the total concentration of the volatiles in the USM (137–141 °C, 4 s) was almost two times higher than that of the W50/T30 (skim milk obtained by preheating at 50 °C, 30 min). Ketones, as a producer of cream and sweetness in the dairy products, were significantly higher in the USM (almost 30 times) than that of the W50/T30. However, the total concentrations of acids in the USM were significantly lower than that of the W50/T30 (50 °C, 30 min). It could be caused by the acid compound (as precursors for methyl ketones, alcohols, and esters [28]), which was produced by lipid oxidation and further induced to produce ketone compounds at high temperatures (137–141 °C, 4 s). 

The aldehydes, ketone compounds, and organic acids were the main products of the Maillard reaction [29,30]. Ketones were the important flavor compounds in the dairy products, which offered a unique flavor and low perception threshold. It could produce an odor of cream and sweetness in the milk samples [17,31]. Alkane compounds could have derived from the automatic oxidation of free fatty acids in milk. Their odor thresholds were high and they had a weak odor in the milk [32]. In this study, ketone and alkane compounds were found almost exclusively in the USM. Therefore, it was deduced that in the presence of fat, the preheating treatment process at 137–141 °C might be more prone to the Maillard reaction, the product of which remains in the skim milk sample. 

In general, the acids from the hydrolysis of triacylglycerols (TAG) were the key flavor components in the milk [19,33]. Hexanoic acid and octanoic acid had oily and waxy aromas, and they were precursors for other flavor compounds, such as methyl ketones, alcohols, and esters [28]. As shown in Table 2, the total concentrations of acids in the USM were about five times higher than those of the RSM and PSM. It could be attributed to the preheating treatment of raw milk before the fat was removed, because studies had reported that the concentrations of lipid-oxidized flavor compounds (hexanal, 2-heptanone, etc.) were very low in the UHT-treated skim milk where fat was removed [12]. In addition, in the presence of fat, the native structure of the MFGM could be damaged by heat treatment, especially under UHT temperatures [14]. The rupture of MFGM could cause the formation of free fat, which was hydrolyzed to produce large amounts of short chain fatty acids at UHT temperatures. These changes could also affect the aroma and flavor of skim milk. 

Aldehydes of the milk were derived from fatty acid metabolism, transamination of amino acids, or Strecker degradation. The hexanal and nonanal were common aldehydes in the milk and had a grassy taste [18]. The total concentrations of aldehydes were not significantly different between the PSM and USM. Additionally, naphthalene and 2-methylnaphthalene were only extracted in the RSM. These compounds were probably transferred from forages [34,35].

### 2.4. PCA and CA of the Volatile Compounds in the RSM, PSM, and USM

Volatile compounds were the important indicator of the flavor quality of skim milk. In our previous study, we did not evaluate the effect of volatile flavor compounds on the flavor quality of skim milk [20]. In this study, the effects of volatile compounds on the flavor of different skim milk samples were further investigated based on the GC-MS data. PCA and CA were used to express the correlation between volatile compounds and the three samples. The results in Figure 5a indicate that the first factor (PC1) presented a positive correlation with hexanoic acid, octanoic acid, dodecane, tetradecane, 2-nonanone, 2-undecanone, 2-ethyl-1-hexanol, and tetrahydro-6-pentyl-2H-pyran-2-one (6-pentyloxan-2-one). The volatiles negatively correlated to PC1 included naphthalene, 2-methylnaphthalene, nonanal, decanal, 2-heptanone, and nonanoic acid. It could deduce that the acids, alkanes, ketones, naphthalene and aldehydes significantly distinguish the flavor of the three skim milk samples. Hexanoic acid, octanoic acid, naphthalene, 2-methylnaphthalene, dodecane, and tetradecane were positively correlated to PC2, whereas nonanal, decanal, 2-nonanone, 2-undecanone, 2-heptanone, nonanoic acid, 2-ethyl-1-hexanol, and tetrahydro-6-pentyl-2H-pyran-2-one were negatively correlated. It showed that the acids, naphthalenes, ketones and aldehydes might distinguish the flavor of the three skim milk samples. From Table 2, we can observe that acids, ketones, and alkanes were mainly found in the PSM and USM, and naphthalene was only found in the RSM. This may be the main reason for the change in aroma and flavor of the skim milk after the preheating treatment.

As shown in Figure 5b, cluster analysis was applied to show the comparability and dissimilarity of the volatile flavor compounds of the three skim milk samples according to the Euclidean distance [36]. The results showed that five clusters were identified, indicating that volatile flavor compounds could provide a better alternative for difference analysis of the three skim milk samples. According to the principle of cluster analysis, the greater the Euclidean distance, the greater the difference between the samples. Therefore, the acids and ketones may be key flavor compounds that differentiate the three skim milk samples. Notably, the concentrations of acids and ketones in the USM were significantly higher than the other two samples (*p* < 0.05, Table 2). In addition, the aldehydes and alkanes had an influence on the flavor difference of the three samples, whereas the other compounds appeared to have no effect on the flavor difference. It could be attributed to the difference in concentrations and the perception threshold.

### 2.5. Relationships Between Sensory Attributes and Volatile Compounds

There were some correlations between flavor compounds and sensory attributes in milk. It could further explain the effect of different treatments on the flavor quality of skim milk. Compared with our previous work [20], the correlations between flavor compounds and sensory attributes were further analyzed in this study. The multivariate data analysis (partial least squares regression, PLSR) was applied to display the correlation between flavor compounds and sensory descriptors [37]. The flavor compounds (X variables) and sensory attributes (Y variables) on the first two factors are presented in Figure 6. The inner and outer ellipse revealed 50% and 100%, respectively, which explained the variance. The attributes “milk flavor”, “milk aroma”, and “caramelized” were correlated with dodecane, tetradecane, 2-ethyl-1-hexanol, 6-pentyloxan-2-one, 2-heptanone, 2-nonanone, and 2-undecanone. It has been reported that 2-heptanone and 2-nonanone were closely related with milk flavor, which possessed an odor of cream and sweetness [17,18,19]. Additionally, the correlation between decanal and “residual mouthfeel” was observed. Notably, the concentrations of these related flavor compounds were significantly the highest in the USM. It was shown that the preheating treatment at 137–141 °C could improve the sensory quality of skim milk. The attributes “fishy” was correlated with naphthalene, 2-methyl naphthalene, and nonanoic acid. It could be the reason for the poor flavor of the RSM. Briefly, although some flavor compounds could not be explained by sensory attributes, the relationship between volatile compounds and sensory attributes was obvious.

## 3. Materials and Methods 

### 3.1. Sample Collection

The raw milk (fat content of 3.3–3.8%) was obtained from Beijing Sanyuan Food Corp., Ltd. (Beijing, China). Milk samples were randomly taken from the same batch and placed in a portable freezer and returned to the laboratory. Subsequently, the samples were stored at 4 °C before using.

### 3.2. Preparation of Skim Milk

The experimental protocol of the skim milk preparation was as follows (as shown in Figure 7). First, raw milk (500 mL) was treated by preheating at pasteurization (85 °C, 15 s) and UHT temperatures (137–141 °C, 4 s) using an experimental tubular sterilizer (DC-UHT-20; Shanghai Dacheng Laboratory Equipment CO. LTD., Shanghai, China), respectively. After a sudden cooling, the raw milk (500 mL) and the samples of preheating treatment were centrifuged to remove fat and obtain three different skim milk samples (RSM, skim milk obtained by centrifugation of raw milk; PSM, skim milk obtained by centrifugation of raw milk after preheating treatment at 85 °C, 15 s; USM, skim milk obtained by centrifugation of raw milk after preheating treatment at 137–141 °C, 4 s). In addition, a MilkoScan TM Minor (FT120, FOSS, Hillerød, Denmark) was used to determine the fat content for the RSM, PSM, and USM. The fat content of each sample was less than 0.5% [6,19,25]. Finally, the three samples were pasteurized (85 °C, 15 s).

### 3.3. Sensory Descriptive Analysis (DA)

The sensory analysis was conducted by using quantitative descriptive analysis [21,38]. Five panelists (1 male and 4 females, aged 25 to 38) were selected by the Department of Food Science and Engineering at Beijing Technology and Business University (Beijing, China). They were trained for sensory analysis of the commercial dairy products at least 30 h [39,40]. During training, sensory properties of the experimental RSM, PSM, and USM were evaluated by the panelists. The sensory lexicon and definitions of skim milk were established (Table 3) according to previous reports [6,41]. The 5-point hedonic scale was used to evaluate preference degrees and the intensities of the key sensory attributes for samples [42] (0–1 = dislike very much; 1–2 = dislike slightly; 2–3 = neither like nor dislike and just about right; 3–4 = like slightly; 4–5 = like very much). Before the assessment, all of the samples were preheated at 37 °C. Sensory evaluation was performed in duplicate (interval of 20 min).

### 3.4. E-Nose Analysis

The RSM, PSM, USM samples were analyzed in six duplicates by the electronic nose (PEN 2, Win Muster Airsense Analytic Inc., Schwerin, Germany) as described previously [27,43]. The PEN2 electronic nose contains 10 different metal-oxide sensors. The array of sensors consisted of the sensor channel, sample channels, and computers. Briefly, samples (10 mL) were placed in a 20 mL vial, which was tightly capped with a polytetrafluoroethylene septum. Then, the sample was stirred magnetically at 40 °C for 5 min, and the headspace (HS) gas was pumped into the sensor chamber via a Teflon tube connected to a needle. During the measurement process, the measurement time lasted for 60 s and the cleaning phase for 300 s. The data were acquired and the computer was used to record the response of the E-nose for every second.

### 3.5. Optimization of HS-SPME Fiber

It has been reported that diverse fibers have different sensitivity, especially when analyzing the same volatile compounds [44]. In this study, four fibers, namely, polydimethylsiloxane (PDMS), divinylbenzene/carboxen/polydimethylsiloxane (DVB/CAR/PDMS), polydimethylsiloxane/divinylbenzene (PDMS/DVB), and carboxen/polydimethylsiloxane (CAR/PDMS), were selected to analyze the volatiles of the milk. The milk sample (8 mL) was added into a 20 mL vial, which was tightly capped with a polytetrafluoroethylene septum, with sodium chloride (2 g) added. The sample was stirred magnetically at 40 °C for 30 min. The SPME needle was inserted in the vial, exposing the fiber to the headspace for 30 min at 50 °C. When the SPME course finished, the fiber was inserted into the GC-MS injector port for desorption for 5 min.

### 3.6. GC-MS Analysis

Volatile compounds of samples were extracted using a solid-phase micro-extraction (SPME) device (Supelco, Bellefonte, PA, USA). All samples (8 mL, in triplicate), 81.6 μg mL-1 2-methyl-3-heptanone in n-hexane [45] (internal standard, 10 μL), and sodium chloride (2 g) were added to a 20 mL vial. Then, the sample was stirred magnetically at 40 °C for 30 min. The SPME needle was inserted in the vial, exposing the fiber to the headspace for 30 min at 50 °C. After 30 min equilibration, the fiber was inserted into the GC-MS injector port for desorption for 5 min.

In the SPME-GC-MS experiment gas chromatograph/mass spectrometer (7890A/5975C, Agilent Technologies Inc., Palo Alto, CA, USA), operating conditions were as follows: the samples were separated by a DB-Wax (30 m × 0.25 mm × 0.25 μm) capillary column (Agilent Technologies, Inc., Palo Alto, CA, USA). Helium was used as the carrier gas with a flow rate of 1.0 mL/min. The injector temperature was 250 °C. Splitless mode was used with a 5 min desorption time. The initial temperature was started at 30 °C, held for 1 min, and then increased at a rate of 5 °C/min to 210 °C and held for 1 min. The ion source temperature was set at 230 °C. The full-scan acquisition was used in the 30–350 amu range. The solvent delay was set at 5 min. 

The volatile compounds were identified using the NIST11 database (Agilent Technologies Inc., Gaithersburg, MD, USA), retention indices (RI), which were calculated by using an alkane series (C6–C30) [46], and the comparison of spectra of authentic standards injected under the same chromatographic conditions. Identified volatile compounds were semi-quantified by comparison with the peak areas of the internal standard.

### 3.7. Statistical Analysis

All data are expressed as the mean ± SD. One-way analysis of variance (ANOVA) and Duncan test were performed using SPSS 22 (SPSS Inc., Chicago, IL, USA). The SPSS 22, Unscrambler^®^ X 10.4 (CAMO Inc., Oslo, Norway) and R version 3.6.0 (MathSoft Inc., Massachusetts, USA) were used for principal component analysis (PCA), cluster analysis (CA), heat map analysis and partial least squares regression (PLSR) analysis. The criterion for statistical significance in all tests was *p* < 0.05.

## 4. Conclusions

Preheating treatment was an important step during the processing of the milk. In this study, we preheated the raw milk at a higher temperature and centrifuged it to obtain a skim milk sample with a satisfactory milk flavor. It revealed that preheating treatment at a higher temperature could change the flavor quality of skim milk. Specifically, the intensity of sensory attributes of the skim milk obtained by centrifugation of raw milk after preheating treatment at 137–141 °C was increased, including milk flavor, milk aroma, caramelized, and sweet taste. These flavors played an important role in the milk products. Additionally, the concentration of the key flavor compounds was significantly higher in the USM than that of the other two samples. It could be attributed to the lipid oxidation, Maillard reaction, and the interaction of lipids with other milk matrices during the preheating treatment in the presence of fat. These findings could provide a novel approach to modifying the sensory attributes and improving the sensory quality of skim milk products.

## Figures and Tables

**Figure 1 molecules-24-02824-f001:**
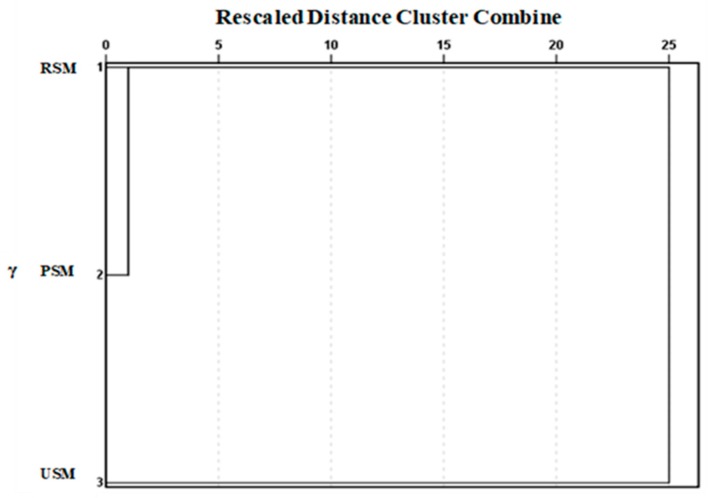
Cluster analysis (CA) of the sensory evaluation for three different skim milk samples. RSM, skim milk obtained by centrifugation of raw milk; PSM, skim milk obtained by centrifugation of raw milk after preheating treatment at 85 °C, 15 s; USM, skim milk obtained by centrifugation of raw milk after preheating treatment at 137–141 °C, 4 s.

**Figure 2 molecules-24-02824-f002:**
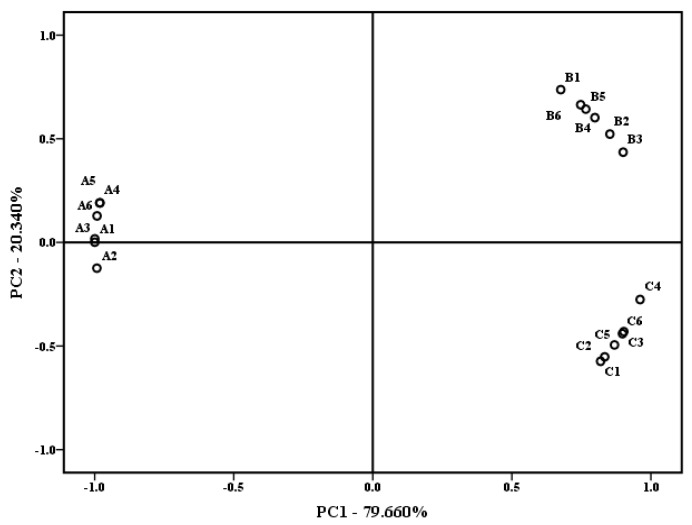
Principal component analysis (PCA) plot for electronic nose data of three kinds of skim milk samples. A, RSM; B, PSM; C, USM.

**Figure 3 molecules-24-02824-f003:**
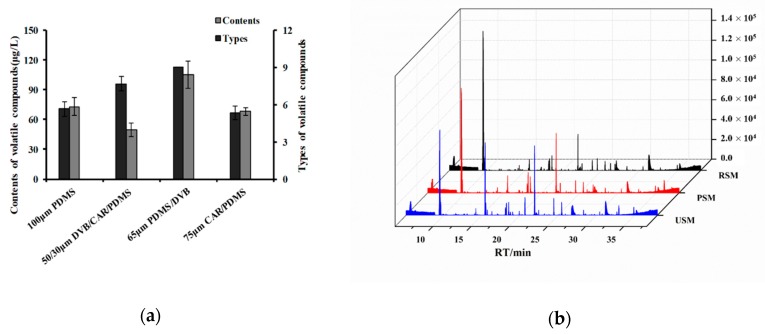
GC-MS analysis of volatile compounds from the milk samples: (**a**) Comparison of amounts and types of volatile compounds from milk extracted by four types of solid-phase microextraction (SPME) fiber (100 μm polydimethylsiloxane (PDMS), 50/30 μm DVB/CAR/PDMS, 65 μm PDMS/divinylbenzene (PDMS/DVB), and 75 μm carboxen/PDMS (CAR/PDMS); all from Supelco Inc., Bellefonte, PA, USA). Means within a group followed by different letters (a–c) are different (*p* < 0.05). (**b**) Total ion currency (TIC) profile of the volatiles from three kinds of skim milk samples.

**Figure 4 molecules-24-02824-f004:**
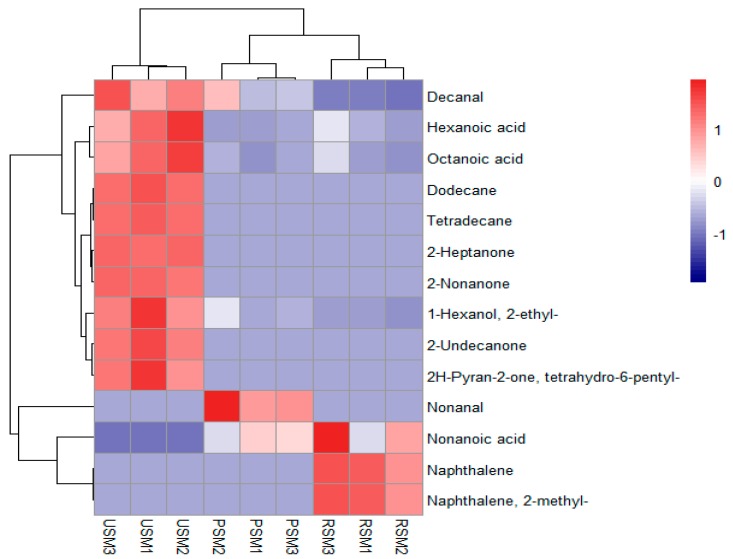
Heat map explains the different concentrations of the volatiles in the three kinds of skim milk samples. (Red represents a high concentration of compounds; white represents a medium concentration of compounds; blue represents a low concentration of compounds.)

**Figure 5 molecules-24-02824-f005:**
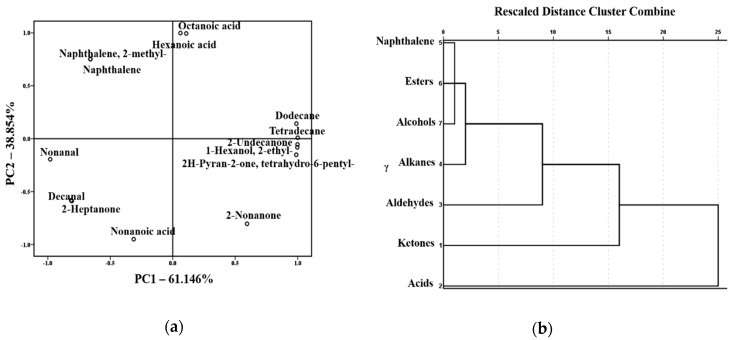
Classification of volatile compounds from the three kinds of skim milk samples: (**a**) Score plot after PCA of the samples. (**b**) Dendrogram from CA of the samples.

**Figure 6 molecules-24-02824-f006:**
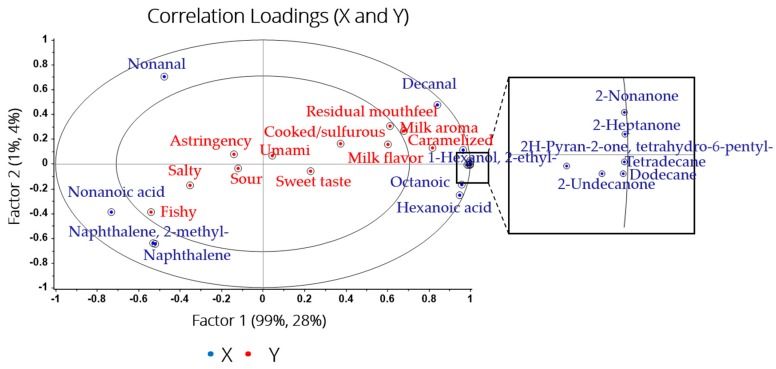
PLSR plot of correlation between volatile compounds and sensory attributes from three kinds of skim milk samples.

**Figure 7 molecules-24-02824-f007:**
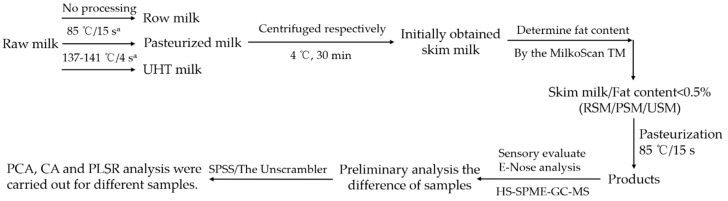
Flow diagram of the preparation and flavor quality analysis of skim milk. (a, Industrialized pasteurization and ultra-high temperature (UHT) sterilization conditions from Beijing Sanyuan Food Corp., Ltd. (Beijing, China)).

**Table 1 molecules-24-02824-t001:** Sensory evaluation results of three different kinds of skim milk samples.

	Sample	RSM	PSM	USM
Descriptor				
Overall liking		1.66 ± 0.44 ^a^	2.50 ± 1.22 ^a^	3.54 ± 0.65 ^b^
Aroma				
Milk aroma		1.66 ± 0.74 ^a^	2.75 ± 0.76 ^b^	3.75 ± 0.53 ^c^
Cooked/sulfurous		2.06 ± 0.86 ^a^	2.35 ± 1.37 ^a^	3.06 ± 0.86 ^a^
Caramelized		0.75 ± 0.65 ^a^	1.28 ± 0.88 ^a^	2.94 ± 0.18 ^b^
Fishy		2.39 ± 0.86 ^a^	0.89 ± 0.60 ^b^	0.45 ± 0.49 ^b^
Flavor/Taste				
Milk flavor		1.60 ± 0.66 ^a^	2.54 ± 0.95 ^b^	3.50 ± 0.93 ^c^
Sweet taste		2.04 ± 1.36 ^a^	2.19 ± 1.00 ^a^	2.63 ± 0.88 ^a^
Umami		1.64 ± 0.58 ^a^	1.80 ± 0.85 ^a^	1.80 ± 1.00 ^a^
Sour		0.38 ± 0.44 ^a^	0.31 ± 0.37 ^a^	0.25 ± 0.38 ^a^
Salty		1.79 ± 1.19 ^a^	1.45 ± 0.68 ^a^	0.95 ± 0.64 ^a^
Astringency		1.00 ± 0.93 ^a^	0.88 ± 1.09 ^a^	0.69 ± 0.53 ^a^
Mouthfeel				
Residual mouthfeel		1.81 ± 0.30 ^a^	2.40 ± 1.03 ^a^	3.29 ± 0.67 ^b^

^a–c^ Significant (*p* < 0.05) difference between samples. RSM, skim milk obtained by centrifugation of raw milk; PSM, skim milk obtained by centrifugation of raw milk after preheating treatment at 85 °C, 15 s; USM, skim milk obtained by centrifugation of raw milk after preheating treatment at 137–141 °C, 4 s.

**Table 2 molecules-24-02824-t002:** HS-SPME-GC-MS analytical results for the volatile compounds of three kinds of skim milk samples.

No.	Compound	RI		CAS	Identified Method	Concentration (μg L^−1^)		
		Cal. ^d^	Ref. ^e^			RSM	PSM	USM
1	2-Heptanone	1180	1184	110-43-0	MS, RI	−	−	13.56 ± 0.22
2	Dodecane	1202	−	112-40-3	MS, RI, STD	−	−	9.51 ± 0.65
3	Tetradecane	1350	1400	629-59-4	MS, RI, STD	−	−	0.84 ± 0.05
4	2-Nonanone	1390	1388	821-55-6	MS, RI	−	0.12 ± 0.04 ^a^	13.84 ± 0.59 ^b^
5	Nonanal	1394	1390	124-19-6	MS, RI	−	0.83 ± 0.27	−
6	1-Hexanol, 2-ethyl-	1494	1484	104-76-7	MS, RI	0.24 ± 0.04 ^a^	0.45 ± 0.17 ^a^	1.49 ± 0.26 ^b^
7	Decanal	1500	1485	112-31-2	MS, RI	0.26 ± 0.06 ^a^	1.34 ± 0.70 ^b^	2.35 ± 0.41 ^c^
8	2-Undecanone	1600	1599	112-12-9	MS, RI	−	−	3.34 ± 0.45
9	Naphthalene	1737	1712	91-20-3	MS, RI	0.28 ± 0.04	−	−
10	Naphthalene, 2-methyl-	1848	1875	90-12-0	MS, RI	0.20 ± 0.03	−	−
11	Hexanoic acid	1880	1866	142-62-1	MS, RI, STD	0.79 ± 0.48 ^a^	0.42 ± 0.09 ^a^	3.92 ± 0.90 ^b^
12	Octanoic acid	2106	2050	124-07-2	MS, RI, STD	1.76 ± 0.76 ^a^	1.70 ± 0.25 ^a^	7.07 ± 1.27 ^b^
13	2H-Pyran-2-one, tetrahydro-6-pentyl-	2194	2179	705-86-2	MS, RI	−	−	2.18 ± 0.42
14	Nonanoic acid	2223	2195	112-05-0	MS, RI	0.55 ± 0.32 ^a^	0.31 ± 0.12 ^a^	−
	Total					4.08 ± 1.59 ^a^	5.05 ± 1.18 ^a^	58.10 ± 2.27 ^b^

^a–c^ Significant (*p* < 0.05) difference between samples. ^d^ Retention indices calculated on DB-WAX column against n-alkanes. ^e^ Retention indices reported by http://webbook.nist.gov/chemistry/cas-ser.html. RSM, skim milk obtained by centrifugation of raw milk; PSM, skim milk obtained by centrifugation of raw milk after preheating treatment at 85 °C, 15 s; USM, skim milk obtained by centrifugation of raw milk after preheating treatment at 137–141 °C, 4 s.

**Table 3 molecules-24-02824-t003:** Sensory descriptors and their definitions for the skim milk.

Descriptor	Definition
Aroma	
Milk aroma	The inherent frankincense of milk, the aroma is gentle, scented, natural, and without a peculiar smell
Cooked/sulfurous	Aromatics associated with cooked milk products
Caramelized	Aromatics associated with caramelized milk products
Fishy	Aromatics associated with fresh fish with skin or canned tuna juice
Flavor/Taste	
Milk flavor	Normal milk taste delicious and slightly sweet, with a unique flavor of pure milk, no other abnormal taste
Sweet taste	Fundamental taste sensation elicited by sugars
Umami	Tastes associated with aginomoto
Sour	Taste of lactic acid
Salty	Fundamental taste sensation elicited by salts
Astringent	Chemical feeling factor producing a dry sensation
Mouthfeel	
Residual mouthfeel	Degree of product left in the mouth after expectoration
Overall liking	Degree of preference for the sample

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
