# Peer review of "Effect of Preheating Treatment before Defatting on the Flavor Quality of Skim Milk"

_molecules, 2019, doi:10.3390/molecules24152824_

Round 1

Reviewer 1 Report

This manuscript continues the authors' previous work on heating of milk. The addition of an electronic nose to the analysis helps with the validation of the human results. One question the authors need to address is whether or not a milk processing plant would be willing to perform a preheating procedure prior to skimming their milk. Such a procedure would add an expense. Another question is what would happen to the milk fat that is recovered after it is removed. Would it still contain enough flavor to make it acceptable for use in butter or another product? 

Author Response

Dear Reviewer:

Thank you for your comments concerning our article entitled “Effect of preheating treatment before defatting on the flavor quality of skim milk”. Those comments are all valuable and very helpful for revising and improving our paper, as well as the important guiding significance to our researches. We have studied comments carefully and have made correction which we hope meet with approval. The responds to your questions are as flowing:

Question 1: The authors need to address is whether or not a milk processing plant would be willing to perform a preheating procedure prior to skimming their milk. Such a procedure would add an expense.

Response: Thank you for your comments. Firstly, with the improvement of the living standard, more and more health-conscious consumers want to have dairy products with high nutritional quality and sensory satisfaction. Importantly, the overall flavor quality remains a primary driver of food choice and overall product value (1. Potts, D. M. et al. Identification of small molecule flavor compounds that contribute to the somatosensory attributes of bovine milk products. Food Chem 2019, 294, 27-34; 2. A Healthy Perspective: Understanding American Food Values. (2017). Retrieved August 6, 2017, from http://www.foodinsight.org/2017-food-and-health-survey). Secondly, with the improvement of the sensory quality of skim milk products, the value of skim milk will also increase. We believe that skim milk with high quality will create higher profits and can make up for the cost of the process. 

Question 2: What would happen to the milk fat that is recovered after it is removed. Would it still contain enough flavor to make it acceptable for use in butter or another product?

Response: Thank you for your comments. Although heat treatment could induce lipid oxidation and produce free fatty acids, methyl ketones, and oxidized fatty acids, it has less effect on the sensory attribute (milkfat, mouthfeel, thickness etc.) of fat itself in milk (Jo, Y. et al. Flavor and flavor chemistry differences among milks processed by high-temperature, short-time pasteurization or ultra-pasteurization. Journal of dairy science 2018, 101, (5), 3812-3828. doi:10.3168/jds.2017-14071). We consider that milk fat after heat treatment can be used to make cream or butter.

We appreciate for Editors/Reviewers’ warm work earnestly, and hope that the correction will meet with approval.

Reviewer 2 Report

The authors carried out an article about the effect of preheating treatments before defatting on the flavor quality of skim milk.

However, the interest of this paper is really now reduced as a similar work has been published in this same journal by four of the 7 authors who signed the present work.  The objective of “demonstrate a better flavor in the skim milk preheated before defatting” is the same, even though instead low temperatures (30-60ºC) in this occasion the authors assay high temperatures (85ºC – 137-141 ºC).

The methodology is nearly the same (Sensory Descriptive Analysis, volatile compounds extraction and GC-MS analysis), although on this occasion it is used E-Nose instead e-Tongue analysis.

The authors avoid to discuss with is own previous results: the authors should compare their two works indicating, for instance, and definitively which would be the best pre-treatment for skimmed milk (50ºC/30 min or 137-141ºC/4 s). The authors should also compare the flavour indicators obtained with milk preheated at 137-141ºC/4, with those obtained in their previous work (milk preheated at 50ºC/30 min).

Author Response

Dear Reviewer: 

    Thank you for your comments concerning our article entitled “Effect of preheating treatment before defatting on the flavor quality of skim milk”. Those comments are all valuable and very helpful for revising and improving our paper, as well as the important guiding significance to our researches. We have studied comments carefully and have made correction which we hope meet with approval. The responds to your questions are as flowing: 

Question 1: The interest of this paper is really now reduced as a similar work has been published in this same journal by four of the 7 authors who signed the present work. The objective of “demonstrate a better flavor in the skim milk preheated before defatting” is the same, even though instead low temperatures (30-60ºC) in this occasion the authors assay high temperatures (85ºC – 137-141 ºC). The methodology is nearly the same (Sensory Descriptive Analysis, volatile compounds extraction and GC-MS analysis), although on this occasion it is used E-Nose instead e-Tongue analysis.

Response: Thank you for your comments. Based on our previous work, the heating temperature was further optimized in this study. Compared with the low temperatures (30-60ºC), the concentrations of the key flavor compounds (2-Heptanone, 2-Nonanone, Octanal, Hexanoic acid etc.) were significantly higher in the skim milk obtained at the high temperatures (137-141 ºC) treatment. The intensity of the main sensory attributes (overall liking, milk aroma, etc.) was closer to the maximum of the hedonic scale. Therefore, we chose a higher temperature for pre-heat treatment to improve the flavor of skim milk. The similar studies as following has been reported that high temperatures could improve the flavor of milk. (1. Licón, C. C et al., Optimization of headspace sorptive extraction for the analysis of volatiles in pressed ewes’ milk cheese. International Dairy Journal 2012, 23, (1), 53-61; 2. Jo, Y. et al., Flavor and flavor chemistry differences among milks processed by high-temperature, short-time pasteurization or ultra-pasteurization. Journal of dairy science 2018, 101, (5), 3812-3828). 

    The complexity of the milk matrix could affect the release of aroma in the milk. The difference in the aroma of skim milk is difficult to distinguish by sensory evaluation personnel. Therefore, E-nose was used to further analyze the aroma in skim milk. Sensory descriptive analysis, volatile compounds extraction, GC-MS, E-Nose and E-Tongue analysis are the most basic analytical tool for evaluating the flavor quality of dairy products. Revised portion were marked in red in the paper (Line 87-89; Line 157-167). 

Question2: The authors avoid to discuss with his own previous results: the authors should compare their two works indicating, for instance, and definitively which would be the best pre-treatment for skimmed milk (50ºC/30 min or 137-141ºC/4s). The authors should also compare the flavor indicators obtained with milk preheated at 137-141ºC/4s, with those obtained in their previous work (milk preheated at 50ºC/30 min). 

Response: Thank you for your comments. As responded in Q1, the condition under 137-141ºC/4s was the best pre-treatment for skimmed milk. At high temperature (137-141ºC/4s), the skim milk had better sensory quality and higher concentration of the key flavor compounds. Specifically, the total concentration of volatiles in the USM (137-141ºC/4s) was almost two times higher than that of the W50/T30 (50ºC/30 min). Ketones, as a producer of cream and sweetness in the dairy products, were significantly higher in the USM than that of the W50/T30. The concentrations of flavor compounds (Octanal, Hexanoic acid, Tetradecane etc.) were also higher in the USM. The related contents were added in the manuscript (Line 87-89; Line 157-167). 

We appreciate for Editors/Reviewers’ warm work earnestly, and hope that the correction will meet with approval.

Reviewer 3 Report

I have carefully read manuscript molecules-550736 entitled „Effect of preheating treatment before defatting on the flavor quality of skim milk“. The scope of the paper is of interest and I have found a general good quality of the research. From experimentation to data evaluation, everything is well organized and clearly described and HS-SPME-GC-MS analysis appears to be carefully performed. In my opinion, this work could be accepted to be published in Molecules after minor revision:

There are many abbreviations without explanation in abstract. Abbreviations must be explained at the first appearance in the text.

Line 17 – please define PSM, UHT and USM

Line 21 – please define RSM

Line 22 – please define PCA. What is E-Nose?

Line 24 – please define PLSR

Line 43 – I think that is not necessary for author to asking themselves…

Line 55 – please define MFGM

Line 68-74 – I think that in the aim of the work, it should be added that the monitoring of compounds that affect milk flavor is done by GC / MS analysis.

Tables are not formatted by the journal rules. It seems that all table are added as figures.

Table 2 – Please, add expected and found RI.

Figure 6 – The image is vague, arrange the font and allocate the names of the parameters to see it better.

Figure 7 – It is better to put original font of Molecules journal rules.

Author Response

Dear Reviewer:    

    Thank you for your comments concerning our article entitled “Effect of preheating treatment before defatting on the flavor quality of skim milk”. Those comments are all valuable and very helpful for revising and improving our paper, as well as the important guiding significance to our researches. We have studied comments carefully and have made correction which we hope meet with approval. The responds to your questions are as flowing:

Question: Line 17 – please define PSM, UHT and USM 

Line 21 – please define RSM 

Line 22 – please define PCA. What is E-Nose? 

Line 24 – please define PLSR 

Line 43 – I think that is not necessary for author to asking themselves… 

Line 55 – please define MFGM

Line 68-74 – I think that in the aim of the work, it should be added that the monitoring of compounds that affect milk flavor is done by GC / MS analysis. 

Tables are not formatted by the journal rules. It seems that all table are added as figures. 

Table 2 – Please, add expected and found RI.

Figure 6 – The image is vague, arrange the font and allocate the names of the parameters to see it better. 

Figure 7 – It is better to put original font of Molecules journal rules. 

Response: Thank you for your comments. All of the above questions were modified in the text. Revised portions were marked in red in the paper. Details as follows: 

Line 17 – please define PSM, UHT and USM (Revised in Line 17-20) 

Line 21 – please define RSM (Revised in Line 23) 

Line 22 – please define PCA. What is E-Nose? (Revised in Line 23-24) 

Line 24 – please define PLSR (Revised in Line 26) 

Line 43 – I think that is not necessary for author to asking themselves… (Deleted in the manuscript)Line 55 – please define MFGM (Revised in Line 58

Line 68-74 – I think that in the aim of the work, it should be added that the monitoring of compounds that affect milk flavor is done by GC / MS analysis. (Added in Line 75-76) 

Tables are not formatted by the journal rules. It seems that all table are added as figures. (Revised in Line 98, 193 and 286) 

Table 2 – Please, add expected and found RI. (Added in Table 2 in Line 193) 

Figure 6 – The image is vague, arrange the font and allocate the names of the parameters to see it better. (Revised in Line 247) 

Figure 7 – It is better to put original font of Molecules journal rules. (Revised in Line 266) 

We appreciate for Editors/Reviewers’ warm work earnestly, and hope that the correction will meet with approval.

Round 2

Reviewer 2 Report

Authors are recommended rewrite the data presented in this paper as a  "Short Communication" as its do not provide substantial amount of new information in comparison with their similar previous article published recently in MOLECULES  (Molecules 2019, 24(9), 1650; https://doi.org/10.3390/molecules24091650 ).

The authors declare: "based on our previous work, the heating temperature was further optimized in this study". Taking into account that the results goes in the same direction as previously work the novelty of the new paper  is really low, althougth the improvement due to the higher heating temperature would be important to be published.

Author Response

Dear Reviewer:

Thank you for your comments concerning our article entitled “Effect of preheating treatment before defatting on the flavor quality of skim milk”. Those comments are all valuable and very helpful for revising and improving our paper, as well as the important guiding significance to our researches. We have studied comments carefully and have made correction which we hope to meet with approval. The responds to your questions are as flowing:

Question: Authors are recommended rewrite the data presented in this paper as a "Short Communication" as its do not provide substantial amount of new information in comparison with their similar previous article published recently in MOLECULES (Molecules 2019, 24(9), 1650; https://doi.org/10.3390/molecules24091650). The authors declare: "based on our previous work, the heating temperature was further optimized in this study". Taking into account that the results goes in the same direction as previously work the novelty of the new paper is really low, although the improvement due to the higher heating temperature would be important to be published.

Response:Thank you for your comments. According to your suggestions, we further discussed the differences between this article and the previous work, and explained the novelty and remark of this article. (Revised in Line 87-89, Line 103-105, Line 113-115, Line 119-121, Line 165-175, Line 214-217, Line 246-249)

We appreciate for Editors/Reviewers’ warm work earnestly, and hope that the correction will meet with approval.